# Investigating the Role of Environmental Factors on the Survival, Stability, and Transmission of SARS-CoV-2, and Their Contribution to COVID-19 Outbreak: A Review

**Leili Mohammadi [1], Ahmad Mehravaran [2], Zahra Derakhshan [3,4], Ehsan Gharehchahi [4], Elza Bontempi [5], Mohammad Golaki [4], Razieh Khaksefidi [4], Mohadeseh Motamed-Jahromi [6], Mahsa Keshtkar [7], Amin Mohammadpour [4,*], Hamid Dahmardeh [8,*] and Samuel Asumadu Sarkodie [9]**

[1] Infectious Diseases and Tropical Medicine Research Center, Research Institute of Cellular and Molecular Sciences in Infectious Diseases, Zahedan University of Medical Sciences, Zahedan 98167-43463, Iran

[2] Department of Parasitology and Mycology, School of Medicine, Infectious Disease and Tropical Medicine Research Center, Resistance Tuberculosis Institute, Zahedan University of Medical Sciences, Zahedan 98167-43463, Iran

[3] Research Center for Health Sciences, Department of Environmental Health, School of Health, Shiraz University of Medical Sciences, Shiraz 71348-14336, Iran

[4] Department of Environmental Health Engineering, School of Public Health, Shiraz University of Medical Sciences, Shiraz 71348-14336, Iran

[5] INSTM and Chemistry for Technologies Laboratory, Department of Mechanical Engineering, University of Brescia, Via Branze 38, 25123 Brescia, Italy

[6] Department of Medical-Surgical Nursing, Nursing School, Fasa University of Medical Sciences, Fasa 74616-86688, Iran

[7] Department of Environmental Health Engineering, School of Health, Hormozgan University of Medical Sciences, Bandar Abbas 79196-93116, Iran

[8] Department of Radiology, Faculty of Medicine, Zahedan University of Medical Sciences, Zahedan 98167-43463, Iran

[9] Nord University Business School (HHN), P.O. Box 1490, 8049 Bodø, Norway

* Correspondence: amohammadpour@sums.ac.ir (A.M.); dr.hamid.dahmardeh@gmail.com or dr.hamid.dahmardeh@zaums.ac.ir (H.D.)

**Abstract:** Studies conducted in the last four years show conflicting findings on the role of the environment in the survival, stability, and transmission of SARS-CoV-2. Based on the current evidence, the factors that affect the severity of COVID-19 include host interaction, environmental stability, virus volume, stability, transmission, social interactions, and restriction measures. Moreover, the persistence of the virus depends on different environmental conditions, videlicet temperature, humidity, pH, salinity, and solar radiation. The outbreak of respiratory viruses is related mainly to temperature and humidity, and geographical locations (latitude). In SARS-CoV-2, mainly temperature and humidity seem to play a fundamental role. Moreover, studies have indicated that social health factors such as equitable health systems, hygiene, and underlying diseases have played a pivotal role in the incidence and outbreak of COVID-19. Therefore, addressing health issues associated with reducing SARS-CoV-2 outbreaks plays an essential role in global health. In contrast, the environmental stimuli of the COVID-19 outbreak are mainly unknown. Given the ongoing threat of the COVID-19 pandemic, it is important to understand the stimuli to respond quickly to emerging SARS-CoV-2 variants while implementing long-term and sustainable control strategies. This review discusses the role of environmental factors and health conditions in the outbreak of SARS-CoV-2.

**Keywords:** environmental health; COVID-19 outbreak; SARS-CoV-2 spread; ambient conditions; waste; SARS-CoV-2 survival and transmission

## 1. Introduction

In December 2019, several cases leading to symptoms similar to those of the known severe acute respiratory syndrome (SARS) were reported in Wuhan, China (i.e., the origin of

the COVID-19 disease). Genetic studies have shown that the causing virus was very similar to coronavirus in bats [1]. The disease outbreak and the mortality rate due to SARS-CoV-2 differed in different countries [2,3]. On the other hand, increased public vaccination has demonstrated that all vaccines are effective against COVID-19-associated hospitalizations and deaths [3,4]. Previous studies showed that more than 567 million people were infected globally by the virus, of which nearly 6 million died [3,5].

Researchers have reported the seasonality effects of the outbreak, which resulted similarly to the common respiratory infections associated with respiratory viruses [6]. Epidemiologists reported the stability of SARS-CoV-2 at temperatures lower than 4 °C, but the virus was found to be sensitive to heat. Several studies have proposed a negative association between COVID-19 mortality and temperature, whereas a positive correlation between mortality rate and daily temperature range (DTR) was suggested [7].

Studies in most countries have recorded the COVID-19 transmission dynamics. Nonetheless, transmission dynamics, disease burden, and death rates vary widely worldwide. Several reasons for the spatial and temporal variation have been observed, such as non-pharmacological interventions (NPI), perception of risk, individuals' behaviors, underlying disease, co-morbidity risk, structural and social determinants of health, and environmental factors [8,9].

A controversial issue concerns the possible relationship between COVID-19 cases and meteorological parameters. Several studies have shown that lower temperatures and humidity are associated with an increased risk of contracting COVID-19 [10]. A study in Saudi Arabia indicated that the daily new COVID-19 cases and mortality rate are influenced by temperature and relative humidity. It has been reported that an average temperature below 3 °C could increase confirmed cases by 4.86%. Another study showed a positive correlation between daily mortality rate due to COVID-19 and temperature range and a negative correlation between daily mortality and relative humidity [11].

The effects of temperature and COVID-19 transmission in 153 counties and 31 Chinese provinces were investigated by Mengyang Liu et al. Their results demonstrated that low temperature positively correlates with daily new COVID-19 cases. In contrast, a negative correlation was found with daily new COVID-19 cases at elevated temperatures [12].

The studies have shown that the spread of the virus in different geographical areas and the occurrence of various pandemic waves can increase the possibilities of outbreak reappearance, while the parametric effect of climate on the spread of the virus has become apparent [13]. The findings increased the expectation that the virus could spread during the winter. Studies in 2021 showed that the transition from summer to winter reduces the temperature and the overall decrease in absolute humidity, hence, possibly increasing the survival of the SARS-CoV-2 [14]. The findings indicate that a temperature gradient negatively impacts the mortality rate. Despite the rise in relative humidity, wind, and precipitation decline, the negative impacts of confirmed new daily COVID-19 cases were apparent. The study reported that assessing spatial and temporal trends of COVID-19 cases requires consideration of seasonality and environmental factors [15].

On the other hand, the COVID-19 outbreak can strongly affect the host's health, depending on existing conditions and access to the healthcare system [16,17]. Thus, attention to environmental parameters is required to predict and reduce the SARS-CoV-2 infection [18]. Accordingly, this study investigates the role of environmental factors and health conditions on the outbreak of SARS-CoV-2 by presenting and discussing the role of parameters such as temperature, humidity, virus volume, and environmental health.

## 2. Materials and Methods

The assessment was conducted in July 2022 while employing databases such as PubMed, ProQuest, Science Direct, and Web of Science to collate articles on coronavirus-related findings. Research on SARS-CoV-2 was performed by examining the clinical manifestations and issues pertaining to the transmission of the virus in different climatic conditions, whereas studies related to environmental health were examined. To collect

information in the field of coronavirus in terms of content, the selected keywords were: "SARS-CoV-2 infections", "environment", "COVID-19 outbreak", "temperature", "relative humidity" "air pollution", "environmental factors", "climatic factors", "municipal waste management", "Hospitalization", "transportation", "Stability", "human health", "underlying diseases", and "economic". A total of 4642 studies were identified through mentioned databases and then 2889 studies remained after the removal of the duplicates. Finally, 227 articles with related titles and abstracts were selected, from which the full texts of 53 articles were reviewed and the relevant content was extracted. Then, a manual reference check was completed and 27 other papers were identified during the selected articles' reference check. In this research, the appropriate content of four reputable sites was also examined as gray literature. The search strategy in databases is reported in the Supplementary Materials.

## 3. Results and Discussion

### 3.1. Coronaviruses: Origins and Pathology

The coronavirus family causes cold-like illnesses in both humans and animals. There are several common coronaviruses between humans and animals, the primary manifestation of which is mild-to-moderate superior respiratory infections, but they rarely have the capability of transmission from animals to humans. Coronaviruses induce infection by attaching to squamous cells and cause pathogenesis. Human coronaviruses can induce problems in the lower part of the lungs, much like pneumonia and bronchitis. However, the novel coronavirus quickly manifested a high mortality rate [19]. SARS-CoV-2 is in the same family as MERS and SARS, and the overall transmission pattern of COVID-19, MERS, and SARS in humans is similar [20].

The most common clinical symptoms at the onset of the disease include fever, cough, fatigue or myalgia, sputum production, and headache. About 42% of patients present with acute respiratory distress syndrome (ARDS), and 61–81% need intensive care [21]. Other studies report different symptoms, including fever, dry cough, muscle aches, fatigue, shortness of breath, loss of taste and smell, and anorexia. Fever and cough were the main symptoms; secondary issues, including kidney damage, liver damage, testicular tissue damage, and diarrhea, have also been reported [22].

Laboratory symptoms include decreased lymphocytes, increased LDH, AST, ALT, blood urea, and creatinine. Most patients had elevated CRP, normal red blood cells, and procalcitonin deposition. Nevertheless, in severe cases, D-dimer increased, and lymphocytes gradually decreased. In fatal cases, the number of neutrophils, D-dimer, blood urea, and creatinine was very high [23,24].

### 3.2. SARS-CoV-2 and Environmental/Climatic Factors

Environmental factors such as humidity, temperature, and latitude affect the virus. Studies worldwide have uncovered that coronaviruses have been pathogenic in the temperature range of 5 °C to 34 °C and the humidity rate of 11 to 85% (Figure 1), following a negative correlation. An increase of 1 °C in temperature and 1% relative humidity has been associated with an estimated lowering of 3% and 1% of new COVID-19 cases, respectively. Overall, temperature and humidity were negatively connected with daily cases, whereas the virus was more contagious in cold and dry climates [25].

Studies on respiratory diseases have demonstrated that seasonal rotation influences the rise of respiratory virus outbreaks [26]. Likewise, the influence of meteorological factors on COVID-19 propagation varies from one season to another. Merow and Urban pointed out a temporary decline in SARS-CoV-2 infection in the summer, whereas, in the autumn, the COVID-19 cases rose and reached the maximum in the winter. A study conducted in Brazil showed the seasonality of COVID-19 transmission [27].

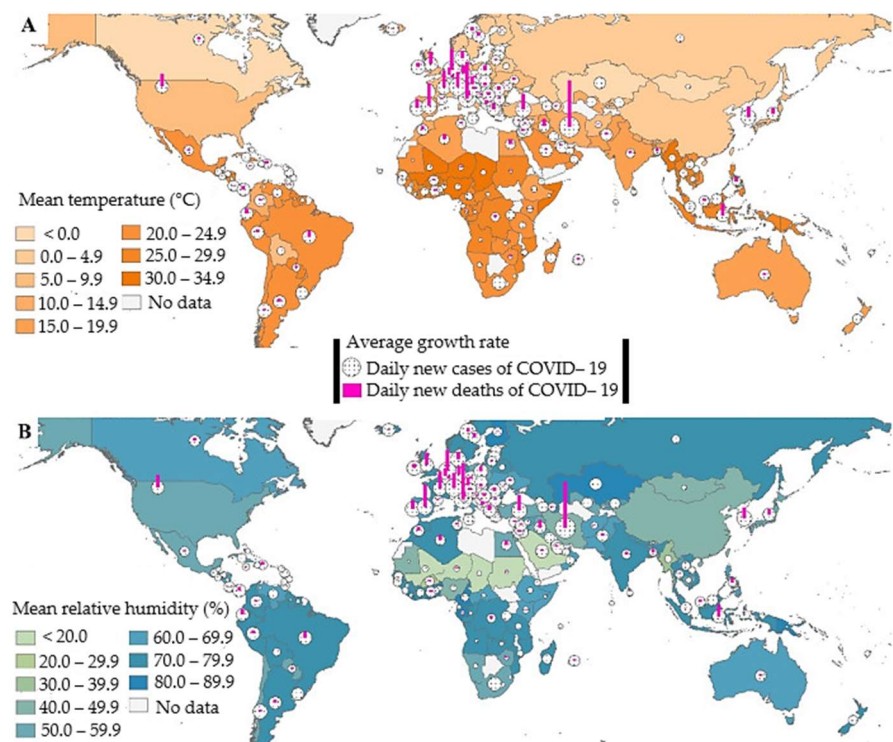

**Figure 1.** (**A**) average temperature, and (**B**) average relative humidity with the average number of new cases and daily deaths of COVID-19. Note: The means were divided daily by the number of the observed days. Reprinted from [25], with permission from Elsevier.

Regarding latitude, some researchers have suggested that temperate at high latitudes has a strong influence on virus spread, such as influenza [28,29]. Few researchers have studied how the spread of COVID-19 correlates with latitude and season, using data from Italy and Egypt [30]. A study with data from 10 countries revealed that COVID-19 infection and its related death rates increased more in the cold than in warm weather [31,32]. Nevertheless, a meta-analysis by Hong-LiLi et al. revealed that season, geospatial scale, and latitude could influence the association between temperature, humidity, and SARS-CoV-2 transmission [33]. Although the virus seemed less infective in the tropics, studies in different countries have shown that the mean diurnal range (52.2%) and temperature seasonality (30.8%) were the most significant determinants of this viral community transmission [34].

In 2020, Bashir et al. studied the possible relationship between environmental health indicators and the COVID-19 pandemic in California. They reported that pollution levels in metropolitan areas (especially in California, which generally has the worst air quality among American states) decreased during quarantine. This reduction in air pollution was due to the reduction in environmental emissions related to halted (major) economic activities, less road traffic, and home quarantine across the United States [35]. Figure 2 shows the decrease in carbon dioxide emissions worldwide between 1945 and 2020 from major historical events. Data provided by the Guardian (Global Carbon Project) show the decline in emissions in 2020 due to the coronavirus pandemic is the biggest in history. The observed reductions could have signaled a behavioral change in moving away from high emission economies.

The results of studies, actions, and field experiences showed the challenges related to environmental health are a very important issue that can seriously affect the severity and duration of the COVID-19 epidemic. Some of the most important environmental health challenges during the epidemic are reported in the subsequent Section 3.2.1, Section 3.2.2, Section 3.2.3, Section 3.2.4.

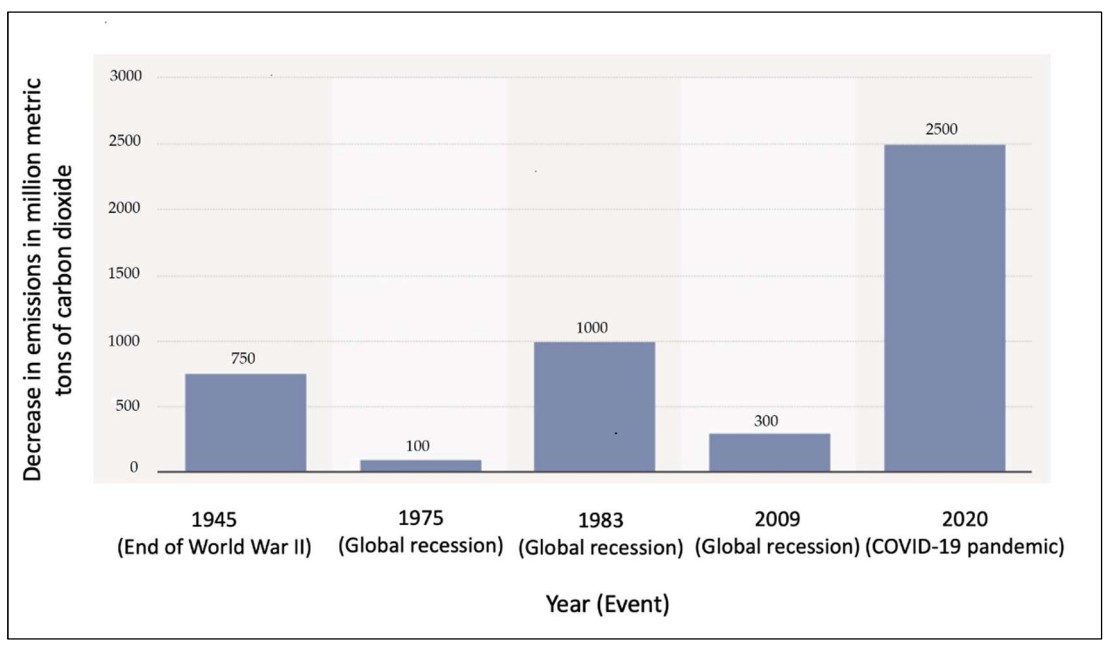

**Figure 2.** Decrease in carbon dioxide emissions worldwide between 1945 and 2020, by major historical event (in million metric tons of carbon dioxide). Data were provided by The Guardian [36].

3.2.1. Disinfectants and Personal Protective Equipment

Structural and facilities readiness, sanitary equipment and disinfectants, adequate and pre-planned personal protective equipment, and sufficient knowledge of health care personnel on how to use them reduce problems related to coordination and executive activities [37]. Therefore, this is among the severe challenges and impacts of the COVID-19 outbreak. There has been a significant increase in medical waste production, with increased use of plastics in health care facilities and trouble with waste management conventional systems functionality, which is fundamental to prevent further population infection and rapid coronavirus outbreaks [38,39]. Moreover, charitable groups acted quickly to provide a large amount of several fundamental materials (such as gloves, masks, and disinfectants) necessary in health facilities. The global water and wastewater treatment market was valued at USD 263.07 billion in 2020. However, the market is projected to reach almost USD 500 billion by 2028 at a CAGR of 7.3 percent in the 2021–2028 period. This growth is expected as the market returns to pre-pandemic levels, having been hit hard since the outbreak of COVID-19 [40].

Among the essential points and severe challenges in managing the COVID-19 epidemic, the implementation of non-expert measures such as unnecessary disinfection of roads was realized in some countries. In several cases, some of these activities were realized, despite their low priorities, and involved the loss of material and human resources, the necessary follow-up was done to stop it, and it was successful [41].

To prevent the spread of SARS-CoV-2, some governments and regulators undertook outdoor spaces and surface disinfection. Nonetheless, the Chinese Center for Disease Control, which initially carried out ample activities to sell equipment to affected countries, later declared that disinfecting outdoor spaces were unnecessary and inefficient in controlling COVID-19. Water bodies, soil, and the atmosphere could be polluted due to the use of disinfectant chemicals. These chemicals could produce potentially toxic and mutagenic secondary products; however, disinfection is necessary for preventing and spreading SARS-CoV-2. All the same, disinfection could be performed with proper precautions to minimize human and natural exposure to detrimental byproducts [42,43].

### 3.2.2. Unknown Transfer Methods

One of the main concerns of the pandemic was the uncertainty of the possible channels of the SARS-CoV-2 transmission [44]. Due to the rapid spread of SARS-CoV-2 compared to similar viruses, the WHO raised concerns about unknown and unapproved methods. One such concern was the potential for the virus to spread through wastewater, particularly medical facility effluents. This issue led to the preparation and promulgation of special instructions to improve the quality of maintenance and the operation of wastewater treatment plants. Among the essential recommendations include, first, increasing the residual chlorine in wastewater treatment plants to at least 0.5 to 1 mg/L. Second, upgrading care programs in water and sewage facilities with regular and permanent control of treatment programs, especially disinfection of water and wastewater. The presence of coronavirus was indeed observed in some samples taken near discharge sites, such as medical facilities for COVID-19 patients. However, the failure to identify the virus in the wastewater plant in some cities, for example, Tehran, showed that wastewater cannot be always considered a suitable place for the presence and transmission of the virus. In addition, in the case of this virus in the sewage, the widespread use of disinfectants in medical facilities and homes appears to guarantee the SARS-CoV-2 inactivation [45]. As a matter of fact, to date, there has been no information on transmission via contact with sewage or polluted water. A few studies on surface water sources and sewage effluents have not detected the infectious form of SARS-CoV-2. Moreover, environmental scientists revealed that SARS-CoV-2 could persist viable for more than 4.3 and 6 days in sewage and water, respectively [46]. Studies on many viruses and recent evidence of the novel coronavirus demonstrate that wastewater-based epidemiology (WBE) is a practical way to evaluate and address viral outbreaks and proves worthwhile for informing related public health policy. Therefore, WBE could help provide public health policies and guidelines [47,48].

Published studies indicate that coronavirus has not yet been detected in tap water, suggesting that municipal water treatment plants and distribution systems can be assumed safe because of the absence of this virus.

### 3.2.3. The Role of Public Transportation in the Spread of COVID-19

The urban transport system necessary for transporting passengers can be a spreading source of the pandemic if basic principles of personal hygiene and safe social distancing (at least one meter) are not observed and maintained [49]. In some cases, the spread of this virus can increase several times compared to the diffusion of several coronaviruses. Indeed, in the urban transport fleet, there are several possibilities of high-contact areas inside buses, city trains, and taxis, as well as in the stations, which can be favorable for the transmission of the SARS-CoV-2 from an infected person to healthy people. In a study by Yayoi Murano et al., the results indicated that restrictions on the domestic transportation system prevented the spread of SARS-CoV-2 in Japan. In Donggyun Ku et al., safe trips in public transport systems were investigated by computing the degree of infection exposure. The consequences demoed that face masking and social distance with a mask throughout rush hours diminished the rate of infection by 93.5 and 98.1% [50].

During urban transport, two groups of people are identified as critical. The first group concerns passengers who do not observe a distance of at least one meter from others and sometimes do not use suitable personal protective equipment. The second group includes drivers (including their families) of urban transports, who, unfortunately, have a high risk of mortality as a result of SARS-CoV-2 infection during the COVID-19 epidemic [51].

In addition to the urban transport fleet, intercity transportation systems, including buses, trains, and airplanes, provide the conditions for spreading SARS-CoV-2 at higher distances than local diffusion. For this reason, great resources were devoted to diffusing information among people and drivers of urban transportation about the most suitable protective equipment, such as gloves and masks. In this frame, masks were imposed for boarding city trains and attending public urban places [51]. In this context, improving the infrastructure for electronic services played an important role in increasing offline activities

and reducing contact by increasing homework. This was recognized as one of the most important actions to reduce the ways of transmission of COVID-19. In addition, this also had an outstanding contribution to reducing environmental pollution. Figure 3 shows the global daily carbon dioxide emissions change by sector during the COVID-19 lockdown by 7 April 2020. With bans imposed on travel, the aviation sector experienced a daily fall of ~60% $CO_2$ emissions by 7 April 2020, relative to annual mean daily emissions in 2019 [52]. It is interesting to note that the lockdown resulted in higher emissions in the residential sector because people spent more time at home.

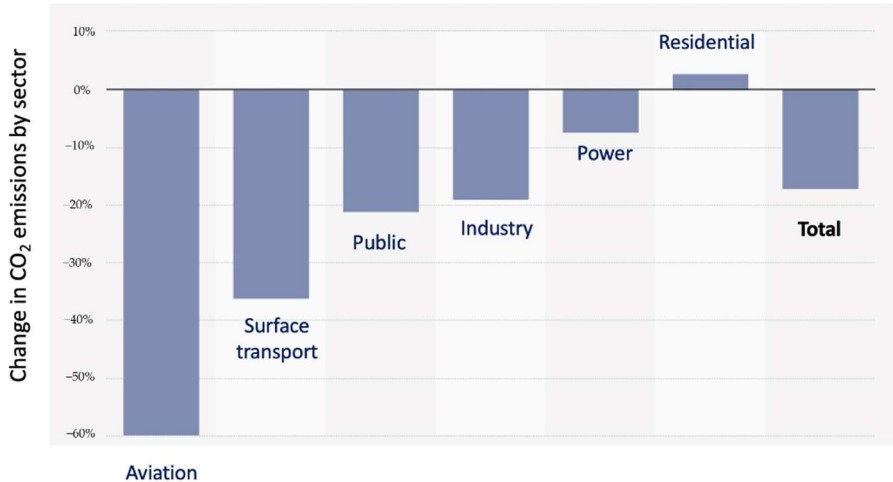

**Figure 3.** Change in global daily $CO_2$ emissions by sector during COVID-19 [53].

### 3.2.4. Hospital and Municipal Waste Management

During the COVID-19 epidemic, indirect side effects have been created, such as increased generation of household solid wastes, especially in health and medical facilities. As a result, the quantity of municipal solid waste (MSW) increased throughout the COVID-19 pandemic; thus, municipal solid waste management faced a significant challenge. Two factors triggered the increase in solid waste throughout the COVID-19 pandemic. The first factor is solid waste generation because of COVID-19 prevention or medication; the second is lifestyle changes, such as home cooking and online purchasing. The limited knowledge about the possible devastating effects of the COVID-19 epidemic on the environment may yield new secondary issues that will be elongated and sometimes more problematic to control and manage [16,54,55].

An estimation indicated that a person infected with COVID-19 could generate approximately 3.4 kg of medical waste daily. Among the severe challenges and impacts of the COVID-19 outbreak, there has been an increase of up to 300% in medical waste production, with an increased use of plastics in health facilities and trouble in waste management conventional systems functionality. Medical waste production in China's Hubei province has increased by 600% (40 to 240 tons), making its management a severe challenge [56]. Nonetheless, an analysis of data from 30 cities showed that solid waste production in major Brazilian cities lowered during the restriction period due to decreased commercial activities [57]. An estimated 8.4 million metric tons of mismanaged plastic waste were generated worldwide during the COVID-19 pandemic, of which 26,000 metric tons were thought to have leaked into the ocean. Asia accounted for almost half of this mismanaged waste, whereas Europe was responsible for a quarter (see Figure 4). The plastic waste mainly originated from sources such as personal protective equipment and COVID-19 test kits.

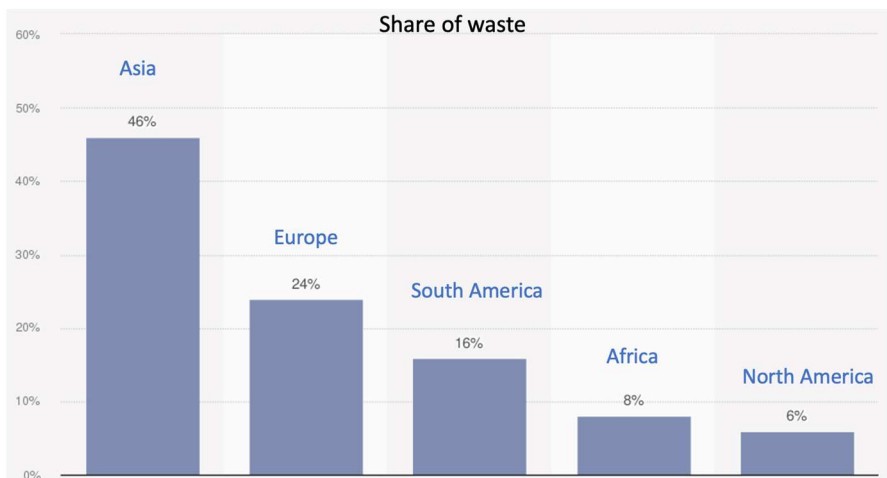

**Figure 4.** Regional distribution of mismanaged plastic waste attributable to the COVID-19 pandemic as of August 2021 (Source: [58]). An estimated 8.4 million metric tons of mismanaged plastic waste was generated worldwide during the COVID-19 pandemic.

A detailed literature analysis shows no evidence of transmission of SARS-CoV-2 through household waste, and the implementation of proper waste management methods along with health requirements and monitoring does not pose the risk of transmission of COVID-19. Because the persistence of the SARS-CoV-2 on surfaces of different inanimate objects (often recyclable) varies accordingly and is short-lived, the number of viruses on the surfaces decreases logarithmically over time. In addition, according to the latest available scientific studies, surface contamination due to pathogens minimizes after 72 h. The effect of high temperatures (e.g., 30 to 40 °C) and humidity (e.g., 50% compared to 30%) can be fundamental for the longevity of viruses [59,60].

### 3.3. SARS-CoV-2 Transmission and Environmental Conditions

In past studies, different pathways of SARS-CoV-2 transmission have been investigated. Most of these studies have observed that the transmission of COVID-19 could occur through inhalation of respiratory droplets or aerosols (especially in enclosed spaces), close contact with an infected person, and high-touching surfaces, human feces, urine aerosols, and cold-chain. Generally, two main routes are known in the transmission of SARS-CoV2 [61]:

1. Direct transmission (transmission through the air by talking, coughing, sneezing, and breathing air droplets)
2. Contact transmission (i.e., through contact with the nasal, oral, and ocular mucosa).

Particles of various sizes have different settling times that Stokes' Law allows to predict precisely. Moreover, many studies have shown that environmental factors significantly impact the transmission of SARS-CoV-2. Prior studies depicted fine particles suspended for hours in the atmosphere. Moreover, fine aerosols less than 5 μm may also contain more viruses than coarse aerosols. Hence, these fine aerosols may play an essential role in SARS-CoV-2 transmission [62–64]. Studies have suggested that direct transmission through inhalation of the infected droplets and contact route is the most common cause of the virus spread [65].

Analysis of virus-infected human specimens has also shown that the transmission of SARS-CoV-2 is not limited to the respiratory tract, and infectious droplets and bodily fluids can easily infect the human conjunctival epithelium. In some cases, it has been suggested that transmission can also occur through the eyes. However, such a possibility is not very common [66]. Studies show that SARS-CoV-2 is generally transmitted mainly through direct or indirect contact with the mucous membranes of the eye, mouth, or nose. Unfortunately, the virus is highly contagious and can spread through asymptomatic patients, so it is

important to isolate patients as soon as possible and adopt severe restrictions on people's mobility [67].

Studies also show a risk of transmitting SARS-CoV-2 infection through contaminated surfaces, although this is more common in closed public places such as public offices, shopping malls, and hospitals. However, affordable information about the virus infectivity through contaminated surfaces in real conditions is unavailable. Finally, the authors acknowledge that scientific data and issues related to the SARS-CoV-2 virus and its potential variants are evolving rapidly. The possible highest-risk-contaminated points and surfaces of SARS-CoV-2 include door handles, public toilets, hospitals or medical centers, factories and industrial zones, public transport, public elevators, traditional/local markets, wastewater plants, and public washrooms [68].

### 3.3.1. Virus Volume and Transmission

Studies have shown that anthropological interventions play a significant role in the transmission of SARS-CoV-2, such that washing the environment and surfaces significantly reduces the volume of the virus. The volume of this virus in the air is an essential factor in its transmission: viral particles in crowded environments tend to aggregate and are transmitted in large numbers, making it necessary to use a mask. Thus, highly populated indoor areas have caused a high number of infections. Nevertheless, studies have shown that masks prevent viruses from entering the airways in those areas in a high percentage [69,70].

### 3.3.2. Stability and Transmission of the Virus

Studies concerning virus detection through polymerase chain reaction (PCR) have shown that viral particles can be detected in fecal samples after the first week of the infection. Other findings suggest that people carry the SARS-CoV-2 in their stool for 27 days or more after being infected, even after recovering. However, transmission through the airway and fecal-oral routes, a reason for great public concern, still needs further investigation and approval. Nevertheless, this remarkable feature of SARS-CoV-2 highlights the need to implement public hygiene measures [71–73].

Evidence suggests that SARS-CoV-2 is most often transmitted through airborne droplets or microdroplets and is abundant in the saliva of infected persons. In a systematic review, Daphne Duvall et al. suggested that COVID-19 airborne transmission could transpire in indoor spaces at long distances. They indicated poor air ventilation was probably a factor in the transmission [74]. The virus first attaches to the surface of the epithelial cells and then enters the cells by the angiotensin-converting enzyme 2 (ACE2), in the same way as all SARS viruses. ACE2 cells are abundant throughout the respiratory tract. Therefore, ACE2 cells are one of the main targets of SARS-CoV-2 infection. However, this issue needs further research [75,76]. According to the guidelines of the WHO, vaccination and health strategies, such as using gloves and superficial antiseptics in public places, reduce the virus risk. This must be coupled with proper ventilation strategies [77].

### 3.3.3. Stability in Different Environments

Coronaviruses may survive from 2 h to 9 days (in airborne particles up to 2 h, on aluminum up to 8 h, on paper up to 4 days, etc., Table 1). At higher temperatures, such as 30 to 40 °C, the virus resistance significantly reduces, and its survival is diminished. SARS is genetically one of the closest viruses to SARS-CoV-2. A decreased lifespan at temperatures above 30 °C has been demonstrated for SARS viruses. Temperatures below 4 °C can increase SARS-CoV-2 survival by up to 28 days. Moderate humidity can make the virus more stable—-it survives more at 50% humidity than 30% humidity, but with higher humidity levels, the stability of the virus decreases. pH is another fundamental factor: coronaviruses have less stability in the environment at pH levels of less than 4 and more than 8 [78].

**Table 1.** SARS-CoV-2 survival rate in the environment [79].

| Duration (h) | Type of Surfaces | Infection Dose * |
|:---:|:---:|:---:|
| 3 | Air | 103.5 to 102.7 TCID50 |
| 72–168 | Plastic | 103.7 to 100.6 TCID50 |
| 8–120 | Steel | 103.7 to 100.6 TCID50 |
| 4 | Copper | Not-reported |
| 24 | Cardboard | Not-reported |
| 72–96 | Paper | Not-reported |
| 48–120 | Wood | Not-reported |
| 96–120 | Glass | Not-reported |
| 01–48 | Cloth | Not-reported |

* Infection dose: The amount of virus necessary to make a person sick.

Since the emergence of SARS-CoV-2, different variants of the virus such as Alpha, Beta, Delta, and Omicron, have yielded a wave of COVID-19 in various countries. In a study, Ryohei Hirose et al. analyzed the environmental persistence of SARS-CoV-2 variants. Their findings showed that the Alpha, Beta, Delta, and Omicron variants survived more than twice as long as the Wuhan variant on plastic surfaces and skin, and the omicron had the longest survival time. The survival time of Wuhan strain, Alpha type, Beta variant, Gamma variant, Delta, and Omicron on the skin were 8.6 h, 19.6 h, 19.1 h, and 11.0, 16.8 h, and 21.1 h, respectively [60].

Virus survival greatly declines in detergent-containing environments. In chlorine-free water at 20 °C, it can survive for up to 2 days, but chlorine-containing waters show much lower virus survival rates. Survival of the virus on surfaces also depends on temperature, humidity, and light conditions. The virus is sensitive to sunlight and survives just for a short time [80,81].

### 3.4. SARS-CoV-2 Importance and Implications for Human Health

3.4.1. COVID-19 and Host Interaction with Virus

Studies have shown that infected people can present different clinical symptoms. For example, many people can have more specific and more robust immune response symptoms after contracting the virus, so they are less likely to develop COVID-19. This issue has also been detected in children with SARS-CoV-2. Children are less likely to have severe inflammatory responses [82,83]. The results also suggest that specific proteins in the blood and lymphocytes could intervene for people with a more robust immune system in response to the virus. In addition, the immune system's role effectively reduces inflammation and lowers the virus's transmission. Those with higher inflammation and severity of the mucous membranes transmit higher volumes of the virus to the environment via the mucus and airborne particles [84,85]. Attention to transmission reduction and quarantine is an important issue that reliably affects the host's interaction with the virus and prevents the continuation of the virus transmission chain.

3.4.2. The Role of Age and Underlying Diseases

The disease mortality rate is related to age. SARS-CoV-2 has caused more deaths in older people. In the United States, SARS-CoV-2 had a 30% mortality rate in people over 85 and a death rate of 27% in people aged 75 to 84 (Table 2). Risk factors and other high-risk underlying diseases include heart failure, diabetes, chronic lung disease, hypertension, cancer, brain disease, kidney disease, liver disease, and people with defective (or suppressed) immune systems. Cardiovascular disease with 10% was the most important risk factor among these diseases, and diabetes was the second risk factor for underlying diseases with 8%. Gender also affected the infection; mortality was higher in men than women [86,87].

**Table 2.** Relationship between disease rate and mortality due to COVID-19 in different age groups of men and women.

| Category | | No. (%) | | |
|---|---|---|---|---|
| | | **Case-Based Surveillance** | | **Supplemental Surveillance** |
| | | **n = 52,166** | | **n = 10,647** |
| Age group (years) | | | | |
| All <65 | | 10,626 | −20.4 | 2681 (25.2) |
| | <18 | 16 (<0.1) | | 5 (<0.1) |
| | 18–44 | 1478 | −2.8 | 423 (4.0) |
| | 45–54 | 2675 | −5.1 | 704 (6.6) |
| | 55–64 | 6457 (12.4) | | 1549 (14.5) |
| All ≥65 | | 41,528 | −79.6 | 7966 (74.8) |
| | 65–74 | 11,245 | −21.6 | 2463 (23.1) |
| | 75–84 | 14,148 | −27.1 | 2900 (27.2) |
| | ≥85 | 16,135 | −30.9 | 2603 (30) |
| | Unknown | 12 (<0.1) | | 0 (0) |
| Sex | | | | |
| Male | | 28,899 | −55.4 | 6449 (60.6) |
| Female | | 22,798 | −43.7 | 4194 (39.4) |
| Other/Unknown | | 469 (0.9) | | 4 (<0.1) |

Blood groups have also been implicated in the evaluation of the factors. People with blood type A are at higher risk, and in people with blood type O, the severity of the disease is very low.

### 3.4.3. The Role of Economic Growth and Access to Public Health

The world economy is devastated by the epidemic [88]. The COVID-19 epidemic is expected to have a strong impact on economic growth, especially in LDCs. In these countries, the decline in external demand, commodity prices, and tourism activities have generally reduced their economic income. However, huge costs were necessary to improve the condition of patients, especially those who need hospitalization and intensive care. Quarantine and other precautionary measures have also severely impacted the economy, whereas anti-cyclical policies are considered insufficient to compensate for the economic shock [57].

In 2019, Iran had the lowest economic growth rate (−9.5%) and the highest inflation rate (35.7%) recorded in the country in the last 20 years. This state of the country's budget has made it impossible to apply adequate prevention, diagnosis, and treatment for COVID-19, and the country cannot introduce similar measures taken in other countries to strengthen strong responses, such as paying public fees for treatment and health [89].

However, the countries with a high number of ICU (specify the meaning) beds available have prevented the loss of human lives compared to others in similar regions with lower availability of these units. For example, Spain has four times more population than Greece; although these two countries had similar infection rates, more ICU beds led to a lower death toll. The EU's unprecedented recovery fund gave some countries (such as Spain) a chance to sustain economic development, innovation, productivity, and resilience [61]. Hence, as the pandemic continues, measures and implementation of health system reinforcement are vital actions to protect the population. For example, in Italy, where the ICU resources were insufficient to face the pandemic, the installation of a considerable number of additional ICUs was realized (see Figure 5) in order to keep up with the increasing number of patients in need of treatment, especially in the region's most hit (for example, Lombardy and Lazio).

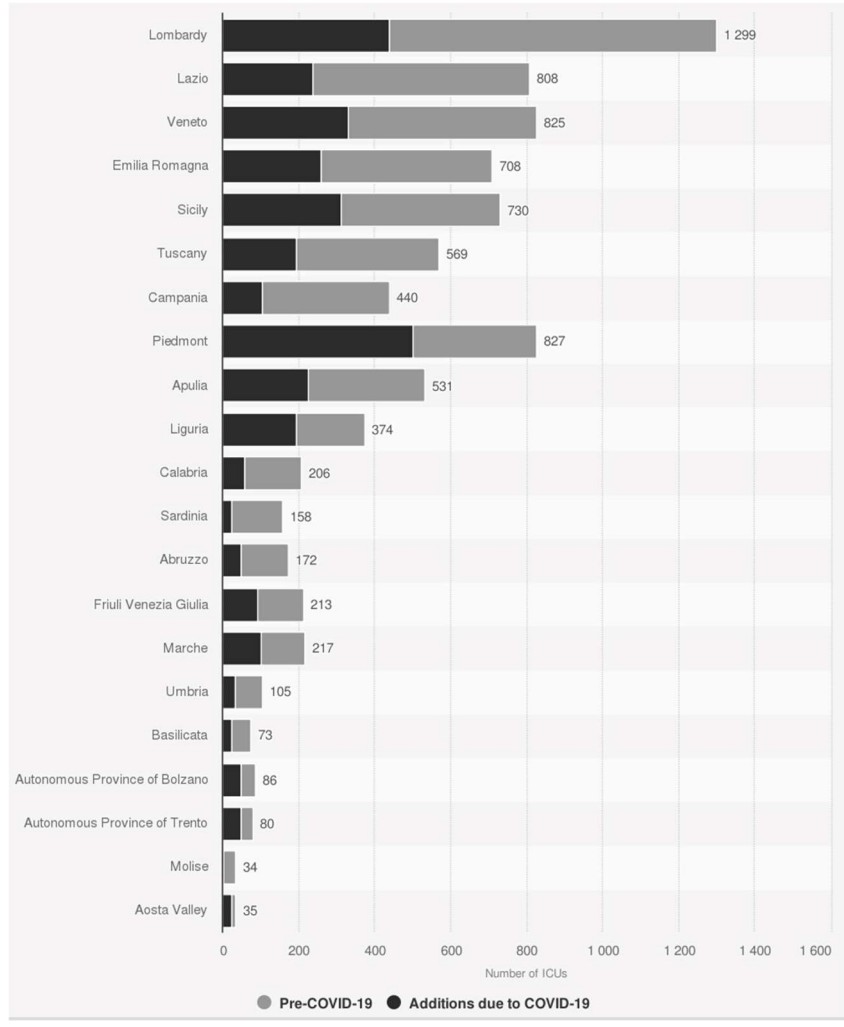

**Figure 5.** Number of intensive care units (ICU) pre- and post-COVID-19 in Italy in 2020, by region. Source [90].

## 4. Conclusions

This review presents findings and discusses the various atmospheric factors, including temperature and humidity that have affected the spread of SARS-CoV-2. Geographic coordinates could impact the relationship between temperature, humidity, and thus, the SARS-CoV-2 transmission. Closing the windows on cold or hot days can affect temperature and humidity. Among environmental factors, the viral load, environmental stability, virus stability, transmission, propagation, and host interaction with SARS-CoV-2 is solely associated with humidity and temperature. Notwithstanding, other factors are not yet determined with high confidence. The stability of the virus in different environments depends on minor changes in temperature, humidity, pH, and solar radiation. Sunlight is one of the factors that seems to inactivate the virus. Health and social factors including age and age-related diseases account for SARS-CoV-2 infection and its outbreak, and the role of economic conditions is fundamental in controlling the virus diffusion in developing countries. Paying attention to health and scientific aspects of reducing SARS-CoV-2 transmission will undoubtedly play an important role in the health of the international community. Today, airborne transmission is considered the main route for COVID-19 transmission, hence, enhancing indoor air quality plays a critical role in preventing the spread of SARS-CoV-2. Nonetheless, limiting close contact and physical distance practice reduces disease transmission indoors and outdoors. Implementing face masks is an essential issue that must be constantly considered. Current evidence has shown no transmission

of SARS-CoV-2 through sewage; however, sewage monitoring can provide indications of potential transmission for more proactive public health responses.

**Supplementary Materials:** The following supporting information can be downloaded at: https://www.mdpi.com/article/10.3390/su141811135/s1, File S1, Investigating the role of environmental factors on the survival, stability, and transmission of SARS-CoV-2, and their contribution to COVID-19 outbreak: a review.

**Author Contributions:** L.M. and A.M. (Ahmad Mehravaran): methodology, writing—original draft, supervision; Z.D. and E.G.: project administration, validation; M.G., R.K. and M.M.-J.: investigation, E.B.: review and editing, supervision; M.K.: investigation, validation; H.D. and A.M. (Amin Mohammadpour): writing—review and editing, supervision; S.A.S.: writing—review and editing. All authors have read and agreed to the published version of the manuscript.

**Funding:** This work was supported by the Research Grant of Zahedan University of Medical Sciences, Iran (Grant No. IR.ZAUMS.REC.1399.382).

**Institutional Review Board Statement:** Not applicable.

**Informed Consent Statement:** Not applicable.

**Data Availability Statement:** The data presented in this research are available on request from the corresponding author.

**Conflicts of Interest:** The authors declare no conflict of interest.

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
