# Peer review of "Investigating the Role of Environmental Factors on the Survival, Stability, and Transmission of SARS-CoV-2, and Their Contribution to COVID-19 Outbreak: A Review"

_sustainability, doi:10.3390/su141811135_

Round 1
Reviewer 1 Report
The present manuscript entitled “Investigating The Role of Environmental Factors on The Survival, Stability, and Transmission of SARS-CoV-2, and Their Contribution to COVID-19 Outbreak: A Review” authored by Leili Mohammadi describes the role of the environment in SARS-CoV-2 survival, stability, and transmission. Furthermore, Studies have indicated that social health factors such as equitable health systems, hygiene, and underlying diseases have played a pivotal role in the incidence and outbreak of COVID-19. Addressing health issues associated with reducing SARS-CoV-2 outbreaks will play an essential role in global health. This review is well written and organized. The review is more a comment than a review in the traditional sense and hence I would strongly advise that it would be published as such. However, certain Minor issues are detailed below to improve the quality of the manuscript.
I advise the authors to take the following points into account while revising their manuscript.
Comment 1: Minor punctuation revision.
Comment 2: The conclusion section needs to be revised.
Comment 3: The homogeneity of the reference section needs to be maintained. In some references, journal names are written in full form and some in abbreviation form. So please check and revise accordingly to the journal's instructions.
Author Response
Dear Reviewer
Thank you for having allowed us to enrich the contents of our research paper through your valuable comments. At the same time, I also record my thanks to the concerned reviewers who commented on our research providing useful guidance to enhance the scientific content of our manuscript. We have considered the comments and suggestions of the reviewers. We believe that all of the comments have been addressed in a way that the reviewers would find satisfactory. We thank the editor and reviewers for their detailed and thoughtful critiques. The manuscript has been greatly improved because of their efforts.
Yours sincerely,
Amin Mohammadpour
Hamid Dahmardeh

Reviewer 2 Report
Please consider using the following studies: After COVID-19: Reorientation of crisis management in crisis; The strategy of vaccination and global pandemic: How framing may thrive on strategy during and after Covid-19; Why some countries win and others lose from the COVID-19 pandemic?: navigating the uncertainty.
However, the authors will decide whether to use the abovementioned publications in this article. Please consider reformatting the abstract. It should be written so that the reader understands the research gap and how the authors have eliminated it. I am asking for accurate proofreading, as I found stylistic and spelling errors in the text. Please provide a separate section about the research methodology and the study results. Now it is mixed. Please clearly indicate your methods following the research problems.
Author Response

(The authors gave the same response as above.)

Reviewer 3 Report
Thank you for the opportunity to read the manuscript entitled "Investigating The Role of Environmental Factors on The Survival, Stability, and Transmission of SARS-CoV-2, and Their Contribution to COVID-19 Outbreak: A Review", and congratulations to the authors for their work.
I highly appreciate the text submitted for review, but here are some suggestions for improvement:
Reviewers suggest changing the structure of the whole manuscript, to adapt the text to the formal sections: Introduction, Materials and Methods, Results, and Discussion (and ending with one brief conclusion). Very importantly is to describe, as independent sections of the others, the Results and Discussion.
The following paragraph (pages 2 and 3 of 16, from line 99 to line 103) should be in the Material and Methods section: "The assessment was conducted in July 2022 while employing databases such as Nature, Pubmed, Medline, NCBI, PsycINFO, Google Scholar, and Google to collate articles on coronavirus-related findings. Research on SARS-CoV-2 was performed by examining the clinical manifestations and issues pertaining to the transmission of the virus in different climatic conditions, whereas studies related to environmental health were examined." The last lines (108 to 110) of that paragraph might be included in the results section.
Perhaps it would be great to provide "Supplementary Materials: The following supporting information can be downloaded at: www.mdpi.com/xxx/s1...". Please, give the aforementioned supplementary materials accompanying the review method section (it may include the search strategy, with the phrase on page 3 -lines 103 to 108-: "To collect information in the field of coronavirus in terms of content, the selected keywords were: “SARS-CoV-2 infections”, “environment”, “COVID-19 outbreak”, “temperature”, “relative humidity” “air pollution”, “environmental factors”, “climatic factors”, “municipal waste management”, “Hospitalization”, “transportation”, “Stability”, “human health”, “underlying diseases”, “economic”.")
Author Response

(The authors gave the same response as above.)

Round 2
Reviewer 2 Report
please replace the incorrect reference in the item. 4 for correct citation. I am giving the correct citation: Dobrowolski, Z. The Strategy of Vaccination and Global Pandemic: How Framing May Thrive on Strategy During and After Covid-19. European Research Studies Journal 2021 14(1), 532-541. https://doi./org/10.35808/ersj/1978
Besides, please review the citation style and standardize it to meet journal requirements.
Author Response
Dear Reviewer
First of all, thank you for your comments and suggestions that allowed us to greatly improve the quality of the manuscript. We agree with all your comments, and we corrected the manuscript accordingly.
Best Regards,

Reviewer 3 Report
Reconsider the length of the conclusion section. It should be triggered to one or two small paragraphs.
Author Response
Dear Reviewer,
First of all, thank you for your comments and suggestions that allowed us to greatly improve the quality of the manuscript. We agree with all your comments, and we corrected the manuscript accordingly.
Best regards,
